# Seismic Fragility Evaluation of Simply Supported Aqueduct Accounting for Water Stop's Leakage Risk

**Zhihua Xiong \***, **Chen Liu, Aijun Zhang \*, Houda Zhu and Jiawen Li**

College of Water Resources and Architectural Engineering, Northwest A&F University, Xianyang 712100, China; liu-chen@nwafu.edu.cn (C.L.); zhuhd2020@nwafu.edu.cn (H.Z.); ljwxncq-686@nwafu.edu.cn (J.L.)
\* Correspondence: zh.xiong@nwsuaf.edu.cn (Z.X.); Zaj@nwsuaf.edu.cn (A.Z.)

**Abstract:** Due to the demands of booming Chinese cities and the increase in urban residents, the safety of aqueduct water transportation structures is noteworthy. A lot of old aqueducts were built in the 1990s and even earlier in the last century and may become vulnerable to potential earthquakes. This paper deals with an evaluation of an aqueduct's seismic vulnerability accounting for leakage risk. Based on the Hua Shigou aqueduct in Ningxia, a probabilistic investigation was carried out to obtain the seismic fragility using Latin hypercube sampling. In the numerical study, the superstructure and substructure of the aqueduct were modeled as beam elements, and the lumped mass method was adopted to simulate the fluid–structure interaction. The rubber water stop's mechanical performance was studied, and its damage states were proposed. Parametric numerical models were then subjected to a set of ground motions according to incremental dynamic analysis (IDA), which contained probabilistic parameters such as water, concrete strength, and bearing performance degradation. Both the system and component levels of the old aqueduct's seismic fragility curves were obtained. It was found that the probability of the water stop's leakage risk is significantly elevated with the increase in ground motion.

**Keywords:** aqueduct; fragility curve; seismic; performance-based; water stop

## 1. Introduction

The South-to-North Water Diversion Project is a national strategic project in China [1]. Its final water transfer scale is estimated to be 44.8 billion m$^3$, which will greatly alleviate the imbalance of water resources between the southern and western areas of China. The Middle Route of the South-to-North Water Diversion Project (MRP), part of the South-to-North Water Diversion Project, is the largest water conveyance project in China and 1432 km in length to transfer [2]. A total of 27 large aqueducts have been constructed in the MRP to cross rivers and valleys.

However, performance-based seismic theory was not considered in the design stage of the aqueduct back in 2000, and it may be more vulnerable to earthquakes. The structure of an aqueduct is very similar to that of a bridge [3]. Aqueducts and bridges have similar super- and substructures. However, compared with a bridge, the coupling effect of fluid in an aqueduct should be considered [4]. Therefore, fluid–structure interaction (FSI) should be taken into account. In terms of the seismic fragility assessment methods of structures in a water tower and liquid storage tank such as the FSI effect, generally speaking, there are two main analysis methods. The first method is the lumped mass method, which is also called the added mass method. In 1957, Housner [5,6] studied the interaction between shaking water and a water tank and proposed the "equivalent concentrated method." He simplified the mass of fluid into two points. The ACI standard [7] was developed based on this study. The lumped mass method was adopted to study dynamic buckling, seismic protection, and the response of elastic soil under horizontal ground excitation of a liquid storage tank in fluid–structure coupling studies [8–11]. However, this method is mainly used in

rigid tanks and cannot evaluate aqueduct sidewalls. The second method is the nonlinear analysis including coupled Lagrangian–Eulerian (CLE), arbitrary Lagrangian–Eulerian (ALE), Smoothed Particle Hydrodynamics (SPH), and so on. D'Alessandro et al. [12] discussed the linear sloshing theory and nonlinear sloshing approach when studying the fluid sloshing coupling simulation of tank vehicle dynamics modeling. Rawat et al. [13] used coupled structural–acoustic (CSA) analysis and CLE to compare the response of water sloshing and hydrodynamic pressure in the two different analysis methods. Zhao et al. [14] applied some simplifications. Masses and shell elements were used to simulate the internal components and equipment of NPP due to its complicated shapes and arrangements. When studying the seismic performance and optimal design of a water tank with an inner ring baffle in a nuclear power plant under seismic load, water and air were simulated by the Eulerian element, while in a shield building, the water tank and ring baffles were simulated by the shell element. Ozdemir et al. [15] used ALE to simulate nonlinear fluid–structure coupling to investigate the application of anchored and unanchored tanks in seismic analysis. In general, the lumped mass method pays more attention to the overall response, while the nonlinear analysis places more emphasis on the local response of the liquid storage tank.

Seismic fragility methods have been widely used in bridge engineering [16]. Moreover, there have been a large number of studies on the seismic fragility of liquid storage tanks, industrial storage pipes, and silos [17–22]. In terms of aqueducts, Valeti et al. [4] studied the seismic response of an elevated aqueduct with a hydrodynamic force and the nonlinear interaction of soil–structure, and the effect of convective masses was found to be significant. Prashar [23] conducted an aqueduct earthquake disaster assessment. Ali Rafiee et al. [24] designed a numerical simulation of a masonry aqueduct under earthquake conditions. Liu et al. [25] studied the shock absorption and seismic effect of rubber support for large aqueducts.

As mentioned above, there has been little investigation on aqueducts' seismic fragility. Because aqueducts are used to transport water, leakage risk should be considered. However, to the author's knowledge, limited research about water stops has been reported. Considering that aqueducts and bridges are similar in structure, this paper carried out an analysis of aqueduct seismic fragility by referring to bridges. The fluid–structure coupling effect in the aqueduct was involved in the analysis model, and the limit state of the water stop deformation was proposed and considered in the seismic vulnerability analysis.

## 2. The Steps of Seismic Fragility Analysis

(1) The appropriate numerical model was built.

(2) According to the fortification intensity and site type of the selected project, the design response spectrum was obtained, which in this study is under Chinese Code GB_51247-2018 [26]. The first nine groups of ground motions were then selected according to the designed response spectrum from the PEER Ground Motion Database [27].

(3) The Latin hypercube sampling (LHS) method was implemented to reduce the correlation between variables. This paper used LHS to extract six groups of three variables.

(4) By running two groups of data analysis and making a comparison of the results, the most vulnerable ground motion direction was identified, and the transverse direction was found to be in the governing ground motion direction.

(5) Each group of models was analyzed with nine groups of ground motion, and the obtained data were extracted and processed.

(6) The damage index of the aqueduct pier and the water stop was proposed and determined.

(7) The fragility curve was drawn, and then the possibility of the exceedance of a limit state under a specific intensity measure was discussed.

## 3. Numerical Modeling

### 3.1. Introduction of the Aqueduct

The water supply project of the north main canal is an important water resource optimal allocation project in Ningxia Province, of which the investigated Hua Shigou aqueduct was part. It is of great significance to improve the ecological environment at the eastern foot of the Helan Mountains, promote the development of characteristic agriculture, and stabilize the industrial and residential water supply in the west of Yinchuan. Hua Shigou aqueduct's total length is 252 m and is divided into 21 spans. The length of each span was 12 m. This study selected a two-span model (shown in Figures 1 and 2). The water transmission project consisted of three sections, among which the north main canal was 47.28 km in total length, including irrigated an area of 25,616.6 acres. Although freezing may occur in extremely low temperatures of Ningxia in winter, some studies have shown that the covered aqueduct will not freeze when the fluid depth is more than 2 m, which we did not cover in this paper.

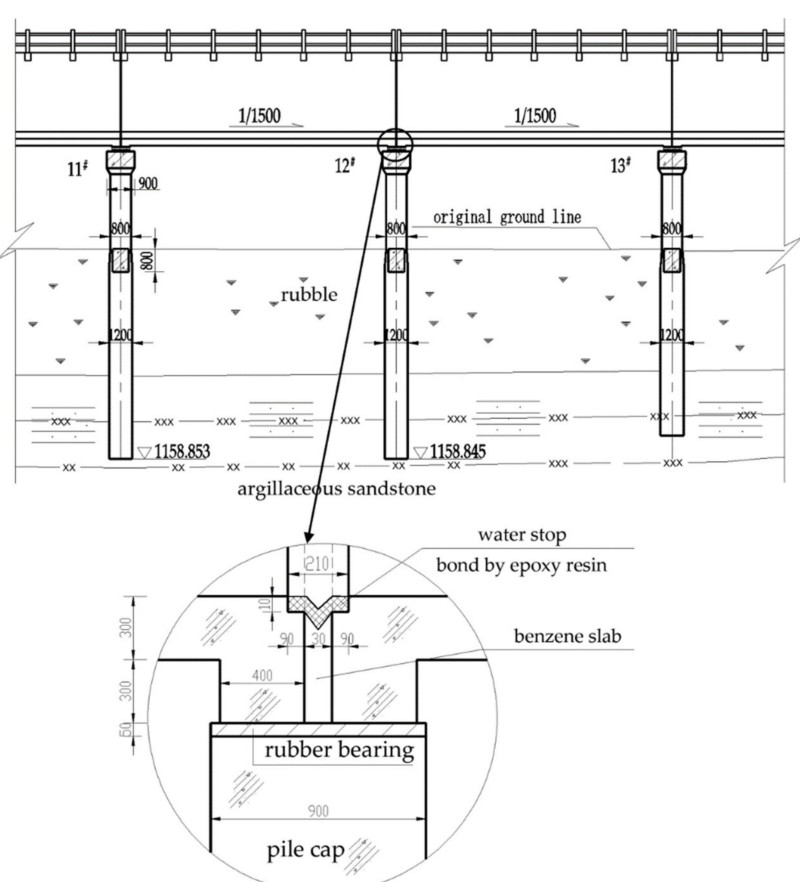

**Figure 1.** Elevation of the aqueduct and the layout of the water stop (unit: mm).

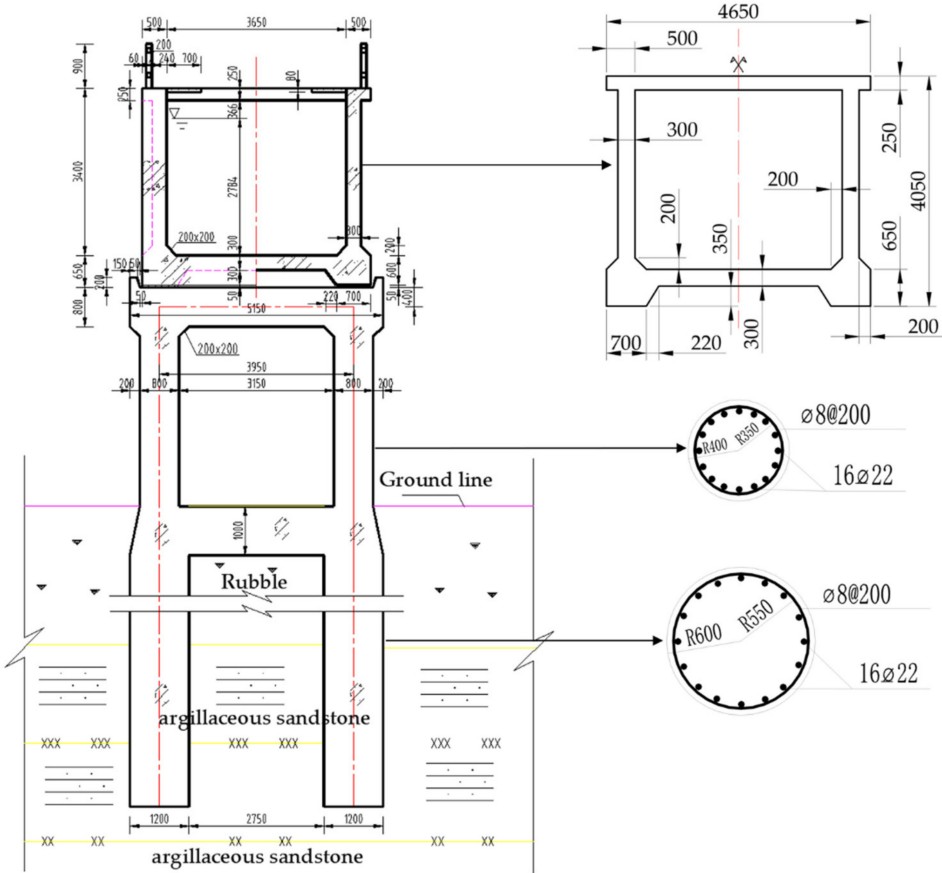

**Figure 2.** Schematic diagram of the aqueduct in the transverse direction (unit: mm).

The water stop is the waterproof structural component between two girders of aqueducts and mainly consists of two types: a rubber water stop and a plastic water stop. A rubber water stop was widely used in the aqueduct, and it can fit the expansion and deformation of concrete in the form of rubber material elasticity. According to structure deformation, water pressure, and waterproof grade, the mechanical property of the water stop was determined.

The geological conditions are slightly dense to medium dense breccia in the surface layer and argillaceous sandstone in the bottom layer. The strata is a nonliquefied soil layer.

### 3.2. Methods of Modeling and Analysis

#### 3.2.1. Lumped Mass Method

FSI is simplified as the lumped mass model according to Housner's theory [6]. The water filled in the aqueducts could be divided into two parts: the impulsive mass and convective mass (shown in Figure 3). For the rectangular tank with a width and water depth of 2 L and H, respectively, the tank with an oscillating water surface is shown in Figure 3a. The equivalent dynamic system is shown in Figure 3b. In Housner's theory [6], the convective mass and impulsive mass are calculated as Equations (2) and (3). M is the total mass in the tank. The height of $M_0$ and $M_1$ are $H_0$ and $H_1$, respectively.

$$M = M_0 + M_1 \tag{1}$$

$$M_0 = M \frac{\tan h(1.7L/H)}{1.7L/H} \tag{2}$$

$$M_1 = M \frac{0.83 \tan h(1.6H/L)}{1.6H/L} \tag{3}$$

$$K_1 = 3 \times \frac{M_1^2}{M} \times \frac{gH}{L^2} \tag{4}$$

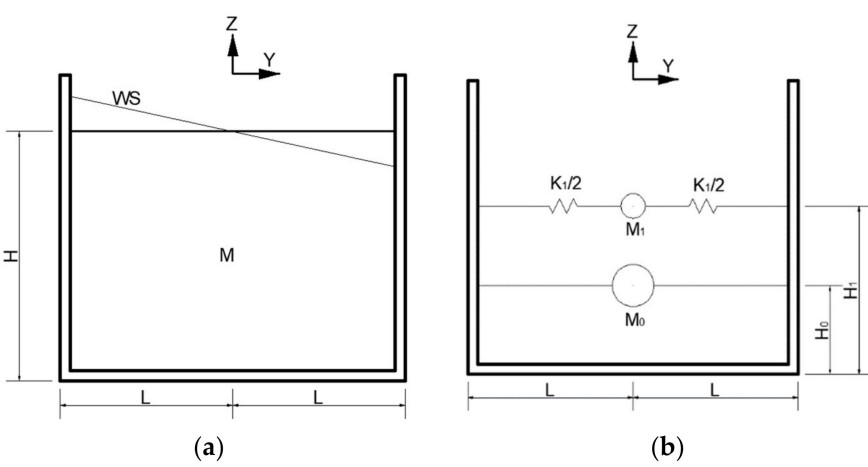

**Figure 3.** (**a**) Sloshing diagram of a tank; (**b**) equivalent dynamic system for a water tank.

In this paper, the connection between the impulsive mass and convective mass was simplified as a connection link with a transverse spring $K_1$.

### 3.2.2. Modeling and Analysis

General FE software SAP2000 was used for incremental dynamic analysis (IDA). The superstructure of the aqueduct and the columns were discrete, as the frame element and pile–soil structural response was simplified as a spring element. The plate rubber bearing was simulated as a multilinear elastic link element for a nonlinear setting [28]. The water stop was simulated by a nonlinear gap element to capture its unique function. The water was considered as lumped mass, as stated previously. The nearby span weight was also modeled as a point load on the column to accurately reflect the aqueduct's dynamic characteristic. The FE model, connecting links, and a model 3D view are illustrated in Figure 4.

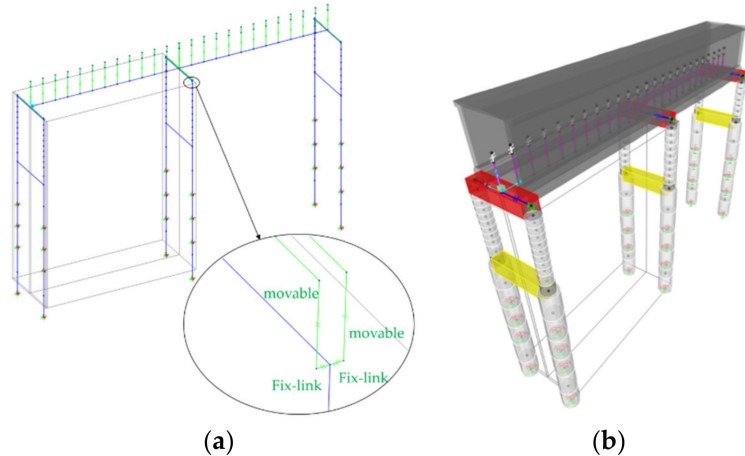

**Figure 4.** (**a**) FE models and connecting links; (**b**) 3D model view.

## 4. Ground Motions and LHS

### 4.1. Selection of Ground Motions

The site condition of the aqueduct is II. The basic seismic peak ground acceleration (PGA) is 0.20 g. The characteristic period of the response spectrum of the basic ground

motion acceleration is 0.45 s. According to the fortification intensity and site type of the project, the design response spectrum was obtained, as shown in Figure 5 [26].

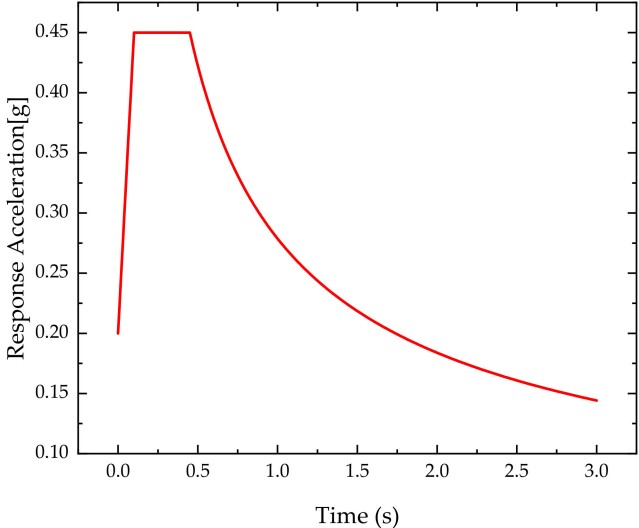

**Figure 5.** The design response spectrum.

The first nine groups of ground motions selected by the utility of the NGA-West2 On-Line Ground-Motion Database Tool are shown in Figure 6.

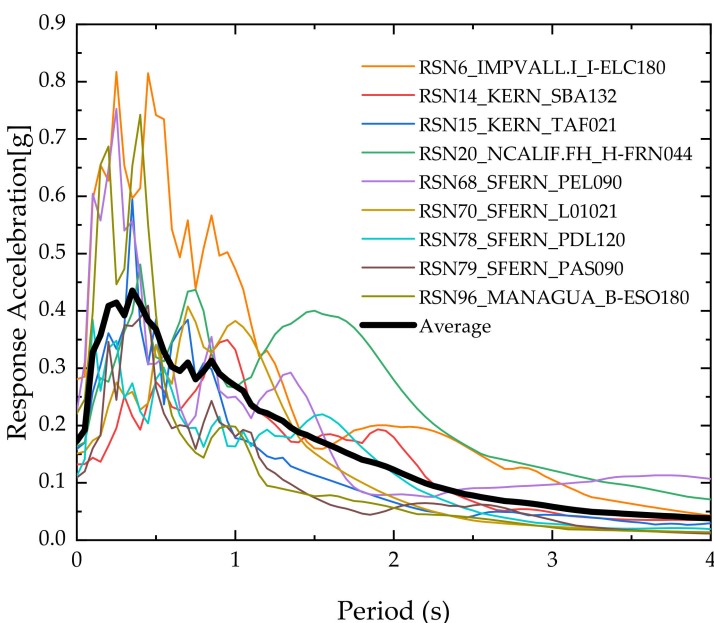

**Figure 6.** The response spectrum from the PEER Ground Motion Database.

In each group, the ground motion with the maximum of PGA was selected and illustrated in Table 1.

**Table 1.** Ground motion records characteristics.

| No. | Earthquake | Station | Year | Magnitude | Horizontal Acc. |
|---|---|---|---|---|---|
| 1 | Imperial Valley-02 | El Centro Array #9 | 1940 | 6.95 | RSN6_IMPVALL.I_I-ELC180 |
| 2 | Kern County | Santa Barbara Courthouse | 1952 | 7.36 | RSN14_KERN_SBA132 |
| 3 | Kern County | Taft Lincoln School | 1952 | 7.36 | RSN15_KERN_TAF021 |
| 4 | Northern Calif-03 | Ferndale City Hall | 1954 | 6.5 | RSN20_NCALIF.FH_H-FRN044 |
| 5 | San Fernando | LA–Hollywood Stor FF | 1971 | 6.61 | RSN68_SFERN_PEL090 |
| 6 | San Fernando | Lake Hughes #1 | 1971 | 6.61 | RSN70_SFERN_L01021 |
| 7 | San Fernando | Palmdale Fire Station | 1971 | 6.61 | RSN78_SFERN_PDL120 |
| 8 | San Fernando | Pasadena–CIT Athenaeum | 1971 | 6.61 | RSN79_SFERN_PAS090 |
| 9 | Managua_Nicaragua-02 | Managua_ ESSO | 1972 | 5.2 | RSN96_MANAGUA_B-ESO180 |

### 4.2. Latin Hypercube Sampling

During the service of the aqueduct, the strength of concrete and the friction coefficient of the rubber bearings are affected by environmental conditions such as freeze-frozen temperature, ultraviolet light, etc. The amount of water is also a variable, mainly due to seasonal variation. In probabilistic theory, the variables are found to be conformed to certain probability distributions. The strength of C30 concrete ($f'_c$) in this study followed a normal distribution of mean = 20.1 MPa and standard deviation = 2.5 MPa [29]. The flow rate conformed to a normal distribution of flow, mean = 10.4 m$^3$/s, and standard deviation = 2.6 m$^3$/s [30]. The weight of water per segment was assumed to follow a normal distribution with a mean = 56.6 kN/m and a standard deviation = 14.1 kN/m. The friction coefficient of rubber bearing is a uniform distribution from 0.02 to 0.05 [31]. These variables are summarized in Table 2.

**Table 2.** Variable distribution of the sampling results.

| Variable | Probability Distribution | μ | σ | Range of Variations |
|---|---|---|---|---|
| Water weight | Normal distribution | 56.6 kN/m | 14.1 kN/m | 0.2–113 kN |
| $f'_c$ | Normal distribution | 20.1 MPa | 2.5 MPa | 15–25 MPa |
| Bearing friction coefficient | Uniform distribution | | | 0.02–0.05 |

Insignificant differences between groups of the same variable would make the theory lack universality. To avoid it, the Latin hypercube sampling (LHS) method was implemented to reduce the correlation of variables between groups. In this paper, six groups of variables were obtained by using LHS, as shown in Table 3.

**Table 3.** Latin hypercube sampling results.

| Group No. | Water Weight (kN) | $f'_c$ (MPa) | Friction Coefficient of Bearing |
|---|---|---|---|
| 1 | 58.97 | 20.7 | 0.0468 |
| 2 | 22.1 | 22.3 | 0.0492 |
| 3 | 83.9 | 16.5 | 0.0395 |
| 4 | 48.9 | 23.9 | 0.0300 |
| 5 | 39.9 | 18.4 | 0.0356 |
| 6 | 86.2 | 20.3 | 0.0221 |

In the scaling approach, all the selected ground motions were incrementally scaled up and down to produce different levels of earthquake intensities. The IDA method was adopted to adjust the amplitude of ground motion Numbers 2–7. The PGA of nine selected groups of ground motion and the amplitude modulation (AM) acceleration are shown in Table 4.

**Table 4.** Ground motion parameter.

| Ground Motion No. | 1 | 2 | 3 | 4 | 5 | 6 | 7 | 8 | 9 |
|---|---|---|---|---|---|---|---|---|---|
| Response acceleration, $g_{max}$ | 0.80 | 0.35 | 0.60 | 0.48 | 0.41 | 0.41 | 0.38 | 0.41 | 0.80 |
| Maximum acceleration, g | 0.28 | 0.13 | 0.16 | 0.16 | 0.22 | 0.15 | 0.11 | 0.11 | 0.22 |
| AM acceleration IDA, g | | 0.39 | 0.32 | 0.48 | 0.66 | 0.75 | 0.77 | | |

## 5. Seismic Fragility Evaluation of Aqueducts

### 5.1. Definition of the Water Stop's Limit State

Water stops are key components of aqueducts. The leakage of the water stop during work will not only affect normal water delivery but also accelerate the corrosion of reinforcement and soften the foundation soil. This will reduce the durability and stability of the structure and seriously endanger the safety of the structure. Especially in the case of a lagged inspection, an earthquake may lead to the aqueduct overturning because the soil around the pier foundation becomes soft. Therefore, it is very necessary to study the vulnerability of a water stop under the action of an earthquake.

Under the action of a transverse earthquake, the shear deformation of the water stop is illustrated in Figure 7.

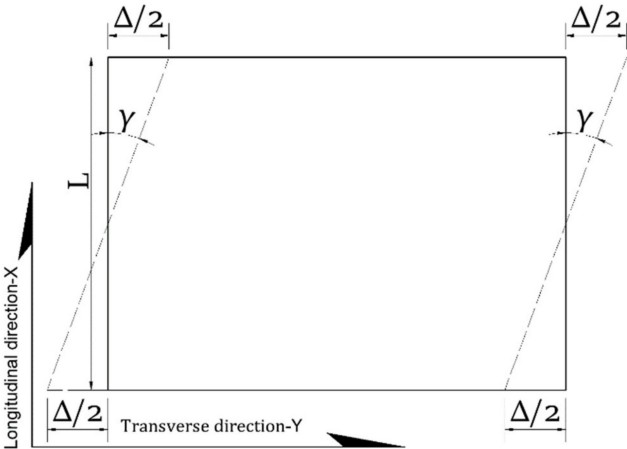

**Figure 7.** Shear deformation of the water stop.

The damage index is defined as Equation (5):

$$\tan \gamma = \frac{\Delta}{L} \tag{5}$$

where $\gamma$ is the rotation of shear deformation. $\Delta$ is the aqueduct transverse relative displacement. L is the width of the connection gap of the water stop.

The rubber water stop is a sealed rubber element, and its shear failure strain is up to more than 300%. Categorizing by function, the water stop's failure modes include relaxation creep failure, stiffness failure, and stability failure. Categorizing by material properties, the water stop's failure modes include fatigue failure, limit failure, and bond failure. Comprehensively considering the bonding, vulcanization process, functionality, and ultimate damage of the water stop, the damage index is generalized in Table 5.

**Table 5.** Definition of different damage states for the water stop.

| Damage Limit State | Damage Index |
| --- | --- |
| Intact | $\tan\gamma < 0.5$ |
| Slight damage | $0.5 \leq \tan\gamma < 1$ |
| Moderate damage | $1 \leq \tan\gamma < 2$ |
| Extensive damage | $2 \leq \tan\gamma < 3$ |
| Complete damage | $\tan\gamma \geq 3$ |

*5.2. Definition of the Pier Limit State*

At present, the damage indexes of bridges are mainly characterized by the curvature or drift rate of the pier. From the angle of bridge engineering, the drift rate of the pier is applicable and accurate during the fragility analysis. Thus, the drift rate of the pier is used as the pier damage index and defined as Equation (6):

$$d_0 = d_c / H \tag{6}$$

where H is the pier height, and $d_c$ is the maximum displacement of the pier top.

The crack of the protective layer of the pier, the yield of the tensile reinforcement, the limit of the tensile reinforcement, and the collapse of the pier column were selected as the boundaries of the damage indices. The damage indices are listed in Table 6.

**Table 6.** Definition of different damage states of the pier.

| Damage Limit State | Damage Index |
| --- | --- |
| Intact | $d_0 < 0.0246\%$ |
| Slight damage | $0.0246\% \leq d_0 < 0.084\%$ |
| Moderate damage | $0.084\% \leq d_0 < 0.09\%$ |
| Extensive damage | $0.09\% \leq d_0 < 0.113\%$ |
| Complete damage | $d_0 \geq 0.113\%$ |

*5.3. Seismic Fragility Evaluation*

Seismic fragility represents the conditional possibility of the exceedance of a limit state under a specific intensity measure. It was expressed as the probability of exceeding a specified performance level for the structure in question by convolving the probability distributions for demand, capacity, and ground motion intensity hazard [32]. The lack of experience in aqueduct damage data, the subjective definition of the limit state, and the advancement of computational tools prove the superiority of this analytical method. It permits predicting the maximum deformation behavior and the collapse risk of an aqueduct under dynamic loadings properly.

To be specific, the exceeding probability $P_f = P(a_i, E_k)$ is defined as the probability that the randomly selected aqueduct samples are in the limit state of $E_k$ under PGA $= a_i$. The conditional probability of a structure exceeding a specific damage level for a given earthquake intensity can be generally expressed as Equation (7).

$$P_f = P\left[\frac{m_d}{m_c} \geq 1\,|\mathrm{IM}\right] = \Phi\left[\frac{\ln(m_d/m_c)}{\sqrt{\beta_d^2 + \beta_c^2}}\right] \tag{7}$$

where $m_d$ and $m_c$ are the demand and capacity of the aqueduct. IM is the PGA (g); $\beta_d$ is the logarithmic standard deviation of structural demand. $\beta_c$ is the logarithmic standard deviation of structural capacity. $\beta_c$ us 0.6 in this paper. $\Phi(x)$ is the standard normal cumulative distribution function. After solving and obtaining x, the corresponding transcendence probability $P_f$ can be obtained by checking the table according to the size of x.

The median engineering demand parameters md and the ground IMs are assumed to follow a logarithmic correlation, as expressed in Equation (8) [33].

$$\ln(m_d) = \ln \alpha + b \ln(IM) \tag{8}$$

where a and b are constant coefficients that can be predicted from the regression analysis.

Compared with the bridge, the water load of the aqueduct lifts the center of gravity of the structure, which theoretically makes the aqueduct structure more susceptible to the transverse force, especially the hydrodynamic influence. Therefore, it was important to ensure whether the horizontal or vertical action was more vulnerable in preanalysis. Itwas found that the model is more vulnerable in the transverse direction after comparing the bending moment of the pier column, which also agreed with the dynamic characteristic of the bridge. A large number of seismic time-history analyses were carried out to investigate the transverse seismic vulnerability of the aqueduct. Using regression parameters, the fitting results are expressed by Equations (9) and (10):

$$\ln(\tan \gamma) = 0.82053 \times \ln(PGA) + 0.205969 \tag{9}$$

$$\ln(d_0) = 0.974 \times \ln(PGA) - 0.5781 \tag{10}$$

Each of the depicted points in Figure 8a shows the logarithmic regression relationship between tanγ and PGA. Each of the depicted points in Figure 8b shows the logarithmic regression relationship between $d_0$ and PGA.

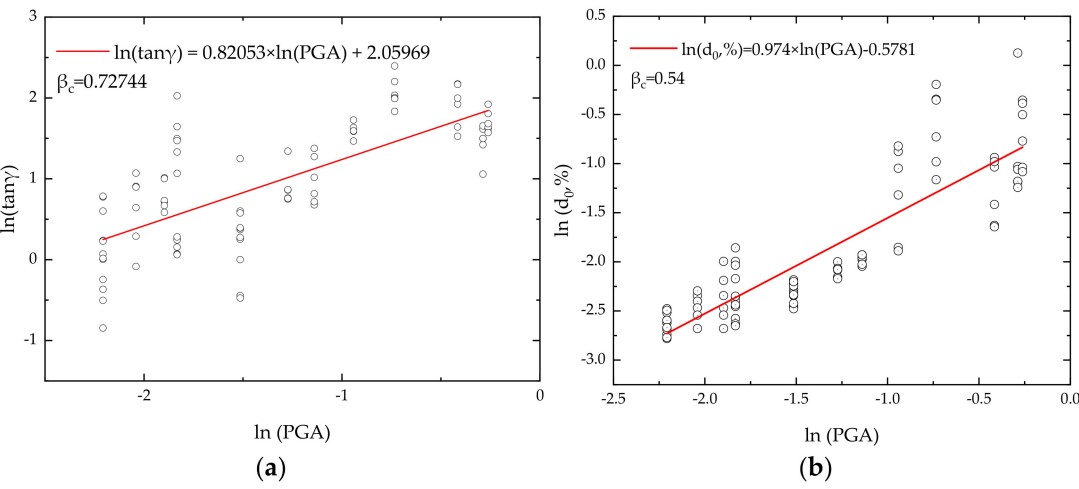

**Figure 8.** Logarithmic regression relationship: (**a**) tanγ and PGA(g); (**b**) $d_0$ and PGA (g).

After that, Equations (11) and (12) can be derived from combining Equations (8)–(10).

$$P_f = P\left[\frac{m_d}{m_c} \geq 1 | PGA\right] = \Phi\left[\frac{\ln\left(\tan Y / \sqrt{\tan Y}\right)}{\beta_C}\right] \\ = \Phi\left[\frac{\ln\left[7.844 \times (PGA)^{0.82053}\sqrt{\tan\gamma}\right]}{0.72744}\right] \tag{11}$$

$$P_f = P\left[\frac{m_d}{m_c} \geq 1 | PGA\right] = \Phi\left[\frac{\ln\left(\tan Y / \sqrt{\tan Y}\right)}{\beta_C}\right] \\ = \Phi\left[\frac{\ln\left[7.844 \times (PGA)^{0.82053}\sqrt{\tan\gamma}\right]}{0.72744}\right] \tag{12}$$

In the end, according to the standard normal distribution table, the fragility curves of the water stop and the pier were drawn, as shown in Figure 9a,b.

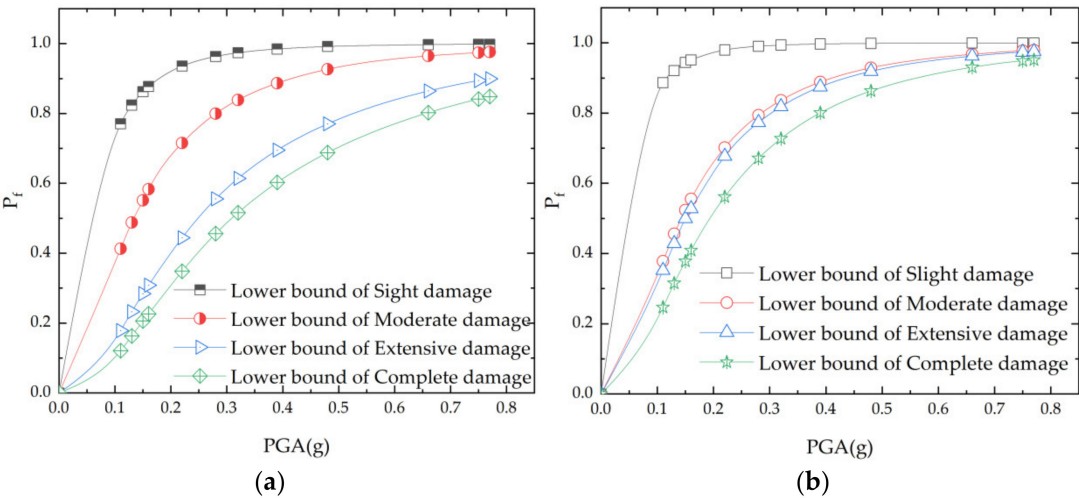

**Figure 9.** (**a**) Fragility curves of the water stop; (**b**) fragility curves of the pier.

## 6. Discussion

It can be seen from Figure 9a that the performance index of the water stop is put forward based on the definition of the shear failure of the rubber bearing. The analysis results show that with the increase in ground motion, the probability of damage to the water stop significantly increases. In the range of PGA < 0.5 g, the water stop is in an intact state, and slight damage is a high probability, while in the range of 0.15–0.3 g, the water stop is prone to be in a slight or moderate damage state. The water stop is gradually transformed to complete damage status when the PGA exceeds 0.3 g and is more than 50% to be completely destroyed. It can be found from Figure 9a that the water stop has good ductility.

Through the analysis of 90 samples, the relationship between the upper and lower limits of the dimensionless drift rate of the pier under the damage limit state and PGA was obtained. The results are shown in Figure 10.

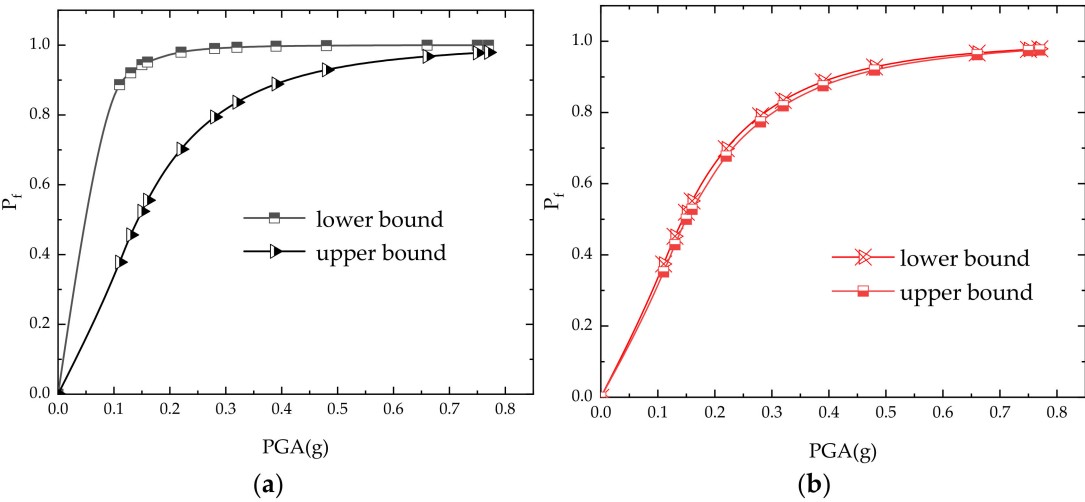

**Figure 10.** *Cont.*

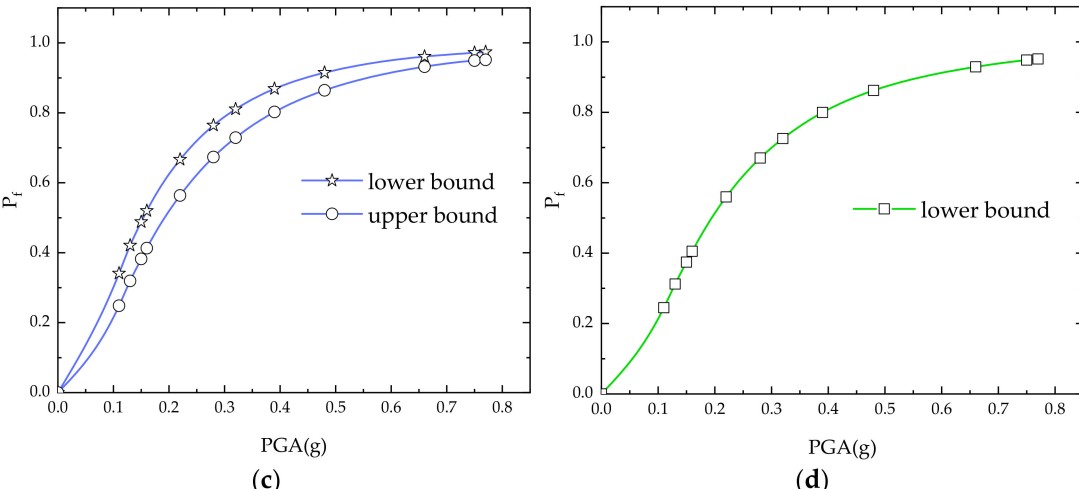

**Figure 10.** Fragility curves of aqueducts: (**a**) slight damage; (**b**) moderate damage; (**c**) extensive damage; (**d**) complete damage.

Comparisons of the fragility curves in terms of the pier are presented in Figure 10. In the range of PGA < 0.15 g, the aqueduct is most likely to be in an intact or slight damage state. In the PGA range of 0.15–0.2g, the aqueduct is most likely to be in a state of slight damage or complete damage, while when the PGA range exceeds 0.2 g, the probability of complete damage is more than 50% and increasing rapidly.

As observed from Figure 11, the aqueduct at present is vulnerable. Seismic rehabilitation is necessary under the current circumstances, and the seismic vulnerability of the water stop is not as significant as the pier. However, the seismic vulnerability of the water stop may be influenced by the performance of the pier and the height difference between adjacent piers.

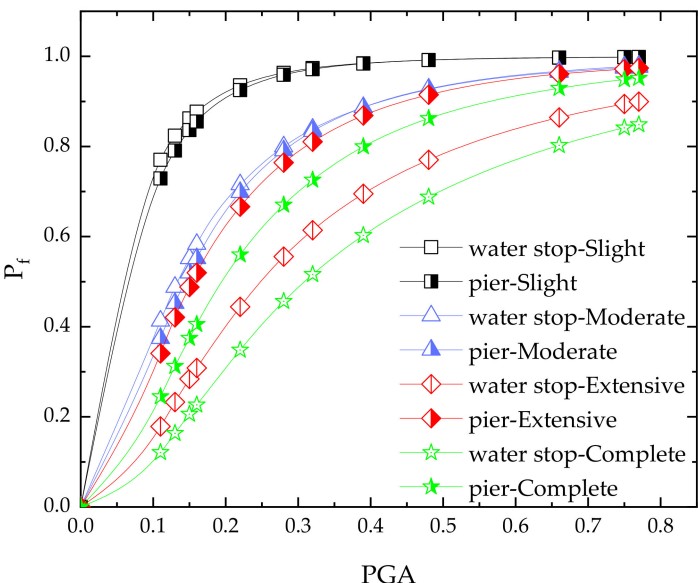

**Figure 11.** Comparison of fragility curves of the pier and the water stop.

## 7. Conclusions

This paper presents a seismic fragility evaluation of an old aqueduct considering leakage risk. Based on the Hua Shigou aqueduct in Ningxia Province, the numerical analysis for risk evaluation was evolved from a probabilistic model, and specific water stop damage states were proposed. By a cluster of IDA, the seismic fragility of the old aqueduct was obtained. The conclusions below are drawn:

1.  An analytical model incorporating water variant, concrete strength, and bearing performance degradation is proposed, in which a water stop is considered a nonlinear gap element. This analytical model can be largely used for the vulnerability analysis of general aqueducts.
2.  The damage state of the rubber water stop is proposed based on the rubber pad theory. The damage limit state of the water stop is defined by the tangent of the shear angle. In the range of $\tan\gamma < 0.5$, the damage status is defined as intact; in the range of 0.5–1, the damage status is defined as slight damage; in the range of 1–2, the damage status is defined as moderate damage; in the range of 2–3, the damage status is defined as extensive damage; and when $\tan\gamma$ exceeds 3, the damage status is defined as complete damage.
3.  With the increase in ground motion, the probability of the water stop's leakage risk is significantly increased. However, the water stop has better ductility compared with the pier according to the vulnerability curve shown in Figure 11. It is necessary to evaluate the seismic fragility of old aqueducts with long service history, and seismic rehabilitation could be needed for these aqueducts based on the vulnerability curve.

The effect of the height difference between adjacent piers and the local damage of prestressed concrete superstructures under fluid–structure coupling will be studied in future work.

**Author Contributions:** Conceptualization, Z.X. and A.Z.; methodology, C.L.; investigation, Z.X. and C.L.; resources, A.Z.; writing—original draft preparation, Z.X. and C.L.; writing—review and editing, Z.X.; visualization, H.Z. and J.L.; supervision, Z.X.; funding acquisition, Z.X. All authors have read and agreed to the published version of the manuscript.

**Funding:** The research was supported by the National Natural Science Foundation of China (Grant No. 51908422) and the Elite Scholar Program of Northwest A&F University (Grant No. Z111022001).

**Institutional Review Board Statement:** Not applicable.

**Informed Consent Statement:** Not applicable.

**Data Availability Statement:** The data presented in this study are available from the corresponding author upon reasonable request.

**Conflicts of Interest:** The authors declare no conflict of interest.

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
