# Peer review of "Seismic Fragility Evaluation of Simply Supported Aqueduct Accounting for Water Stop’s Leakage Risk"

_water, doi:10.3390/w13101404_

Round 1

Reviewer 1 Report

From the viewpoints of damage of water stop and pier, the method for evaluating the seismic fragility of aqueduct is explained.

This method may be useful for judging the necessity of restoring the old aqueducts.

It will be necessary  to check following points again:

  1. Page 4, line 131 : What is the meaning of "total quality" ?  Is it same as "total mass"?
  2. Page 5, Figure 3(b) : What is the meaning of H1 and H2 ?
  3. Page 8, Table 2. : What is the meaning of f'? Is it the strength of  concrete ?
  4. Relationships between the damage limit state and the damage index are  shown in Tables 4 and 5 for water stop and pier. Are these universal ? 

Author Response

We appreciate the reviewer’s comments. The reply and revisions according to the comments are described below:

  1. Page 4,line 130: “total quality” is corrected as “total mass”;
  2. Page 5, Figure 3(b): the meaning of ‘’H0’’ and ‘’H1’’ are added;
  3. Page 8, line 169 : the meaning of f'c is added;
  4. The damage limit state and index are universal.

Reviewer 2 Report

The authors reported the study on seismic fragility evaluation of simply supported aqueduct ac- 2 counting for water stop’s leakage risk. In general, the main conclusions presented in the paper supported by the figures and supporting text, and therefore it is recommended that the manuscript can be published in Water. However, to meet the journal quality standards, the following major comments need to be addressed

Specific major comments and suggestions:

  1. Abstract can be polished. It first should contain problem formulation and precise significant outcome of the result. Also authors should elaborate the general applicability of the study.

  1. Introduction writing part should be be improved. The writing and presentation of the introduction lacks in clarity. The paper requires a significant amount of rewriting to clarify all aspects of it, especially the novelty and new findings of this work that need to be clearly mentioned.

The authors have mentioned “As mentioned above, there has been little investigation on aqueduct’s seismic fragility. Considering that aqueduct and bridge are similar in structure, this paper has carried  out analysis of aqueduct seismic fragility by referring to bridge.”…if this is the novelty  of the current work,  this point need to be elaborated.

  1. Page 2: “…. were selected according to the designed response spectrum from PEER Ground Motion Database - PEER Center.”—please explain. Add reference if necessary.
  2. Authors add "methods" chapter the way so anybody can repeat the computational procedures, like a recipe.
  3. ” FSI is simplified as lumped mass model according to Housner theory”.. please explain. Add reference if necessary
  4. “At present, the damage indexes of bridges are mainly characterized by the curvature 215 or drift rate of support and pier. Since the pier of aqueduct is not very high, drift rate of 216 pier top can be used as pier damage index and defined as Eq. (6):” ….The reader not quite understand why this is important in the present study ?

  1. Results and discussion: - The paper is overall descriptive can be improved. Comparison with existing literature can be improved, limitations of the results should be discussed, future research should be outlined.

  1. The motivation for the study has to be mentioned in detail. What are the implications for this study in actual applications?

  1. Typographical errors: There are several minor grammatical errors and incorrect sentence structures. Please run this through a spell checker.

Author Response

We appreciate the reviewer’s comments. The reply and revisions according to the comments are described below:

  1. The abstract is polished and some conclusions are added.
  2. The introduction and the novelty of the present work are elaborated and revised in the section Introduction.
  3. Reference about PEER is added.
  4. “Methods” has been contained as the second section: The Steps of Seismic Fragility Analysis
  5. Reference [6] is added.
  6. Relevant explanations have been added in line 221.
  7. Future research is outlined in line 317.
  8. Regarding to the motivation, it is clarified in line 9-12.
  9. An overall spell check is made and some minor mistakes are corrected.

Reviewer 3 Report

In equations 1 -3 are missing literature citations -Equivalent dynamic system for a water tank (Housner, 1963)  [6] in article. No info about ratio H/L. On line 297 is "This analytic model" - should be analytical model? Generally an article well prepared, Aqueduct is tested by engineering method, but lack of info what is new in this research. 

Author Response

We appreciate the reviewer’s comments. The reply and revisions according to the comments are described below:

  1. The equations [1-3] are labelled with references in line 133.
  2. The info about ratio H/L can be calculate by Figure 2, Figure 3 and Table 3.
  3. "This analytic model" is revised as "This analytical model" in line 304.
  4. As mentioned in the literature review in Introduction, limited reports have been found about the research of aqueduct, which we try to cite the relevant articles as much as we can in the references.
